# Distribution of naturally -occurring NS5B resistance-associated substitutions in Egyptian patients with chronic Hepatitis C

Hala Rady Ahmed[1], Nancy G. F. M. Waly[1], Rehab Mahmoud Abd El-Baky[1,2]*, Ramadan Yahia[2], Helal F. Hetta[3,4], Amr M. Elsayed[5], Reham Ali Ibrahem[1]

1 Department of Microbiology and Immunology, Faculty of Pharmacy, Minia University, Minia, Egypt,
2 Department of Microbiology and Immunology, Faculty of Pharmacy, Deraya University, Minia, Egypt,
3 Department of Medical Microbiology and Immunology, Faculty of Medicine, Assiut University, Assiut, Egypt,
4 Department of Medical Microbiology and Immunology, Faculty of Medicine, Merit University, Sohag, Egypt,
5 Tropical Medicine and Gastroenterology, Faculty of Medicine, Minia University, Minia, Egypt

* rehab.mahmoud@mu.edu.eg, dr_rehab010@yahoo.com

## Abstract

### Background

NS5B polymerase inhibitors represent the cornerstone of the present treatment of Hepatitis C virus infection (HCV). Naturally occurring substitution mutations to NS5B inhibitors have been recorded. The current study intended to demonstrate possible natural direct acting antiviral (DAA)—mutations of the HCV NS5B region in HCV patients in Minia governorate, Egypt.

### Methods

Samples were collected from 27 treatment-naïve HCV patients and 8 non-responders. Out of 27 treatment-naïve patients, 17 NS5B sequences (amino acids 221–345) from treatment-naïve patients and one sample of non-responders were successfully amplified. Nucleotide sequences have been aligned, translated into amino acids, and compared to drug resistance mutations reported in the literature.

### Results

NS5B amino acid sequence analysis ensures several novel NS5B mutations existence (more than 40 substitution mutations) that have not been previously documented to be correlated with a resistant phenotype. It was found that K304R (82.4%), E327D and P300T (76.5% each) substitutions were the most distributed in the tested samples, respectively. S282T, the major resistance mutation that induces high sofosbuvir-resistance level in addition to other reported mutations (L320F/C) and (C316Y/N) were not recognized. Q309R mutation is a ribavirin-associated resistance, which was recognized in one strain (5.9%) of genotype 1g sequences. Besides, one substitution mutation (E237G) was identified in the successfully amplified non-responder sample.

**Data Availability Statement:** All relevant data are within the manuscript and its Supporting Information files.

**Funding:** The author(s) received no specific funding for this work.

**Competing interests:** The authors have declared that no competing interests exist.

## Conclusion

Our study showed various combinations of mutations in the analyzed NS5B genes which could enhance the possibility of therapy failure in patients administered regimens including multiple DAA.

## 1. Introduction

Hepatitis C virus (HCV) is a viral pandemic, expresses the most prevalent blood-borne viral infection that preferentially replicates in the liver [1, 2]. It is the main reason for chronic viral hepatitis that might lead to critical liver fibrosis, cirrhosis, and hepatocellular carcinoma (HCC) [3, 4]. Global HCV predominance varies around the world [5]. Egypt reports the highest HCV predominance worldwide [6–8]. Overall, the epidemiology of HCV is evolved because of a scale-up in screening and prevention actions and elevated cure rates in the time of interferon-free DAA therapy [9]. HCV is marked by genetic heterogeneity, with at least 7 major genotypes are classified [10]. In Egypt, genotype 4 is the predominant genotype, comprising 90% of the infected Egyptians [11–13]. However, GT-4 is a very heterogeneous genotype with high genetic diversity and more subtypes compared with other genotypes. Nowadays, 18 subtypes of HCV-4 have been identified, with 4a as the dominant subtype [14, 15].

HCV is a positive-sensed single-stranded RNA virus that belongs to the Flaviviridae family. The genome of HCV involves 9000 to 9800 nucleotides which encode structural and non-structural proteins. The viral particle forms of structural proteins, core and the envelope glycoproteins E1 and E2. The structural proteins separate apart from the non-structural proteins by the short membrane peptide p7 [16–18].

NS2, NS3, NS4A, NS4B, NS5A and NS5B are the non-structural proteins which included in viral replication and viral polyprotein processing. NS5B functions as RNA-dependent RNA (RdRp) polymerase, synthesizing RNA using an RNA template [19, 20]. In 2011, the Directly Acting Antivirals (DAA) appeared as a chief development in HCV therapy. These drugs targeted the viral proteins involved in viral replication [21]. DAAs treat HCV infections with a smaller management period, elevated cure rates, and fewer side effects. Amongst DAA drugs, sofosbuvir is a nucleotide analog HCV NS5B polymerase inhibitor, as the National Committee for the Control of Viral Hepatitis adopted which was verified to be an effective treatment with SVR rates approaching 95% and used as orally (60 mg once per day) [22–25].

Resistance Associated Substitutions (RASs) are natural substitutions that account for different degrees of resistance to DAAs and pose a great challenge to the effectiveness of HCV antiviral therapy [26, 27]. The RASs can happen naturally in patients infected with HCV before starting DAA treatment [28]. In all studies, Sofosbuvir (SOF) revealed a great genetic barrier. However, limited substitutions with resistance to SOF were noticed in NS5B [29–31].

In this study, we demonstrated possible natural direct-acting antiviral (DAA)—mutations of the HCV NS5B region in HCV patients in Minia governorate, Egypt.

## 2. Patients and methods

### 2.1 Patients and sample collection

Blood specimens were collected between January 2019 and August 2019, 27 from randomly selected treatment-naïve HCV infected patients and 8 non-responders at Viral Hepatitis Management Center, Minia governorate, Upper Egypt, Egypt. Specimens were collected as part of

the routine work of the central laboratory. The study protocol conformed to the ethical guidelines of the 1975, Declaration of Helsinki, as revealed in *a priori* approval (No. 65/2019) by commission of the ethics of scientific research, Faculty of Pharmacy, Minia university. The baseline HCV RNA levels were assessed before the initiation of HCV antiviral therapy and the results were available as the routine workup for diagnosis and management of HCV infection in patients' records. The HCV-RNA was amplified on Stratagene Mx3000P system using Artus HCV QS-RGQ-PCR Kit. Amplification followed by simultaneous detection was carried out using RT-PCR, with TaqMan assay for a particular region of the HCV genome. The lower limit of detection for the assay is 30 IU/ml.

## 2.2 Viral RNA extraction and cDNA synthesis

Approximately, 3 ml of blood were collected from each patient, the samples were shipped cooled (4˚C) within 6 hours, centrifuged at 1500 rpm, and sera were separated then distributed in aliquots, stored at -20˚C.

Viral RNA was collected from patients' sera using QIAamp DSP virus Kit (cat#60704, Qiagen, Germany) following the manufacturer's instructions. The concentrations of nucleic acid were determined by NanoDrop™ 1000 Spectrophotometer V3.7 (Thermo Fisher Scientific Inc, Wilmington, DE, USA). Then after RNA extraction the eluted RNA was stored at -80˚C until further use. The cDNA strand was synthesized from the extracted HCV RNA using High-Capacity cDNA Reverse Transcription Kit (cat#4368814, Applied Biosystems, USA).

## 2.3 Amplification of NS5B gene by Nested Polymerase Chain Reactions (Nested PCR)

The primers utilized for NS5B amplifications and DNA sequencing are JA230, 5`-TACCAT CATGGCTAA(A/G)AA(C/T)-GAGGT (outer sense, 8008–8032); JA233,5`-ATGATGTTA TGAGCTCCA(A/G)GTC(A/G)TA (outer antisense, 8663–8687); JA231, 5`TATGA(C/T) ACCCGCTG(C/T)TTTGAC (inner sense, 8256–8276); and JA232, 5`-CCTGGTCATAGC CTCCGTGAA (inner antisense, 8616–8636) [32]**.** These primers were utilized in a standard single-tube RT-outer PCR with 1.5 mM MgCl2. The temperature was optimized at 42˚C for 60 min, then by 95˚C for 5 min, then thermo-cycling for 30 cycles at 95˚C for 30 s, 50˚C for 30 s and 72˚C for 30 s, with a final extension step at 72˚C for 5 min. The inner PCR was done under the same thermo-cycling conditions, utilizing the inner sense primers.

PCR negative samples were extracted again using The MagMAX Viral RNA Isolation Kit for viral nucleic acid isolation which used to deliver yields of high quality RNA and DNA with little sample-to-sample variation as 35–75% of the input RNA should be recovered. The Mag-MAX™ magnetic bead technology is useful for concentrating RNA from diluted samples. Then, the PCR tried a second time but gave no bands.

## 2.4 Purification of PCR product and sequencing

Purification of PCR products was performed using QIAquick PCR Purification Kit (250) (cat#28106, Qiagen, Germany). The purified NS5B PCR products were sequenced utilizing the Big Dye Terminator V 3.1 Cycle Sequencing Kit (cat#4337455, applied biosystems, USA) on the automated sequencer genetic analyzers.

## 2.5 Analysis of NS5B partial gene sequences

The NS5B gene sequences were analyzed for genotyping, amino acid sequence identification and mutations utilizing the data and tools available at the Genafor- Open Services for Medical

Research Website (https://www.genafor.org/index.php). A phylogenetic tree was performed based on the assessment of the HCV NS5B sequences with reference strains. The MEGA 6 software was used for the phylogenetic analysis. The reported sequences through this study have been submitted to the GenBank database under accession MN794403, MN894516, MN794406, MN794404, MN794405, MN794407, MN794408, MN794409, MN794410, MN794411, MN794412, MN894517, MN794413, MN794414, MN794415, MN794416, MN794417 and MW307936.

## 3. Results

Clinical and virological features of patients involved in this study are summarized in Table 1. The current study involved 35 patients (27 treatment-naïve HCV patients and 8 non-responders (who failed to attain an early viral response associating combined sofosbuvir (400mg/day), daclatasvir (60mg/day) for 6 months therapy)). Out of 27 treatment-naïve patients, 17 were male and 10 were female with ages ranged between 23 and 75 years old. For non-responders, 7 patients were male, and one patient was female.

Regarding the baseline features of the tested treatment-naïve patients, it was observed that ALT and Platelet counts displayed abnormal level in 8/27 and 4/27 patients, respectively. Declined levels of WBCs count and prothrombin activity were shown in 3/27 and 5/27 patients, respectively. On the other hand, 8/27 patients displayed abnormal hemoglobin level (Table 1). For non-responders, it was detected that 5/8 patients displayed normal levels of all experienced variables. In addition, variable clinical characteristics were observed in 3/8 patients.

In this study, the success rate of amplification for NS5B fragments was 17/27 of the collected treatment-naïve patients' samples and one sample of the collected non-responders samples that could be attributed to the difficulty to amplify NS5B fragments, low viral load and RNA level in most of the unsuccessfully amplified samples, requirement of extremely trained professionals and longer turnaround time [32–34].

Most treatment-naïve patients in our study were genotyped as HCV genotype-4a 13/17 (76.5%), Subtype 3a and 4l were both identified in one sample (5.9%), and subtype 1g was recognized in 2/17 samples (11.7%). The only amplified sample of the non-responders was genotype 4o. To study substitutions along NS5B protein, Egyptian HCV amino acid sequences were aligned to the NS5B world consensus sequences. The relationship amongst the sequences isolates was shown in the phylogenetic tree (Fig 1). In Egypt, the major NS5B polymerase inhibitor [nucleoside inhibitor (NI)] available is SOF. So, we mainly examined the positions having resistance associated variants (RAVs) to SOF. Other positions, 368, 411, 414, 448 and 553 correlated to another NS5B polymerase inhibitor resistance, Dasabuvir (non-nucleoside inhibitor (NNI)) were not included in this study.

The significant S282T mutation (accompanying extreme level of resistance to SOF) as well as other mutations that are formerly documented with resistance to SOF (L320F/C) and (C316Y/N) were not detected in the successfully amplified specimens. while various novel NS5B mutations that have not been formerly announced to be correlated with a resistant phenotype were determined. As 5 amino acid substitutions (K304R, E327D, P300T, R270K and N333S/R/A) were set to be the most abundant RASs in the treatment-naïve patients. Only one substitution mutation (E237G) was recognized in the successfully amplified specimen of non-responders (Table 2).

Mutations associated with ribavirin resistance were examined at positions D244N, Q309R, and A333E of NS5B. The Q309R mutation was recognized in one strain (5.6%) of genotype 1g. In this study, mutations of D244N and A333E were not found in any sequence, However

**Table 1. Clinical characteristics of the tested treatment-naïve HCV patients and non-responders.**

| ID | Age (years) | Gender | ALT Up To 40U/L | Platelet count X10³ (150–400) | PC % (70–110%) | Viral load X 10⁵ (IU/ml) | WBCs Count 10³/mm³ (4000–11000) | Hg g/dL female:12–14 male:13–15 |
|---|---|---|---|---|---|---|---|---|
| A. Treatment-naïve patients | | | | | | | | |
| 1 | 75 | M | 23 | 166 | 75 | 19 | 7.2 | 14.2 |
| 2 | 53 | M | 49 | 250 | 97 | 15.8 | 4.2 | 11 |
| 3 | 60 | F | 55 | 115 | 65 | 5.5 | 5.8 | 14.5 |
| 4 | 50 | F | 36 | 210 | 78 | 1.82 | 8.1 | 12.1 |
| 5 | 23 | M | 29 | 219 | 78 | 14.8 | 8.4 | 14.9 |
| 6 | 58 | M | 118 | 168 | 76 | 11 | 5.7 | 14.8 |
| 7 | 70 | M | 28 | 201 | 84 | 1.12 | 5.4 | 17.3 |
| 8 | 58 | M | 22 | 174 | 87 | 64.9 | 5.1 | 12.9 |
| 9 | 29 | F | 29 | 207 | 97 | 0.41 | 5.9 | 11.4 |
| 10 | 58 | M | 31 | 148 | 64 | 10.3 | 6.3 | 13.3 |
| 11 | 64 | M | 48 | 177 | 88 | 5.2 | 3.9 | 13.5 |
| 12 | 67 | M | 27 | 156 | 85 | 12.6 | 6.8 | 14.7 |
| 13 | 55 | F | 141 | 154 | 76 | 2.2 | 5.2 | 13.4 |
| 14 | 74 | M | 136 | 185 | 97 | 5.63 | 5.2 | 16.8 |
| 15 | 65 | M | 29 | 134 | 69 | 14.1 | 5.4 | 15.1 |
| 16 | 34 | M | 25 | 247 | 80 | 5.45 | 4.1 | 13 |
| 17 | 32 | M | 28 | 210 | 90 | 2.94 | 6.1 | 15.9 |
| 18 | 58 | M | 105 | 172 | 75 | 10.3 | 6.7 | 14.1 |
| 19 | 56 | M | 36 | 186 | 80 | 0.55 | 5.9 | 14.4 |
| 20 | 69 | M | 37 | 175 | 77.5 | 0.753 | 4.2 | 13.5 |
| 21 | 38 | M | 67 | 215 | 90 | 1.07 | 6.3 | 16.2 |
| 22 | 26 | F | 26 | 176 | 75 | 0.93 | 5.6 | 11.5 |
| 23 | 33 | F | 38 | 125 | 75 | 1.03 | 4.9 | 11.6 |
| 24 | 62 | F | 34 | 184 | 82 | 0.895 | 5.6 | 12.2 |
| 25 | 54 | F | 36 | 105 | 65 | 10.5 | 3.4 | 11.7 |
| 26 | 55 | F | 32 | 168 | 75 | 0.791 | 4.8 | 11.2 |
| 27 | 56 | F | 38 | 183 | 77.5 | 0.697 | 3.9 | 11.8 |
| B. Non-responders: | | | | | | | | |
| 28 | 57 | M | 56 | 115 | 62 | 1.75 | 5.2 | 15.1 |
| 29 | 27 | M | 26 | 220 | 87 | 0.72 | 6.2 | 14.8 |
| 30 | 47 | M | 31 | 195 | 82 | 0.98 | 7.3 | 14.5 |
| 31 | 53 | M | 29 | 176 | 80 | 0.87 | 5.8 | 13.9 |
| 32 | 52 | M | 23 | 216 | 77.5 | 0.64 | 5.7 | 13.2 |
| 33 | 63 | M | 25 | 135 | 62 | 12 | 3.8 | 12.8 |
| 34 | 73 | M | 33 | 165 | 75 | 0.91 | 4.7 | 13.1 |
| 35 | 49 | F | 93 | 92 | 55 | 1.1 | 3.5 | 10.8 |

**ALT: alanine aminotransferase, WBCs: White blood cells count, PC: Prothrombine Concentration, Hg: hemoglobin**

N333S/R/A and G333A polymorphisms were recognized in 10 (58.8%) and 1 (5.8%) of 17 treatment-naïve specimens' sequences. Other mutations, D310N and T329I which are associated with peg-IFN/RBV resistance were not detected in any of the examined sequences (Table 2). The presence of combinations of many substitutions in one sample was found in the tested treatment-naïve patients. As 5 to 10 substitution mutations was the most prevalent (47%) among the tested samples of the treatment-naïve patients followed by the presence of

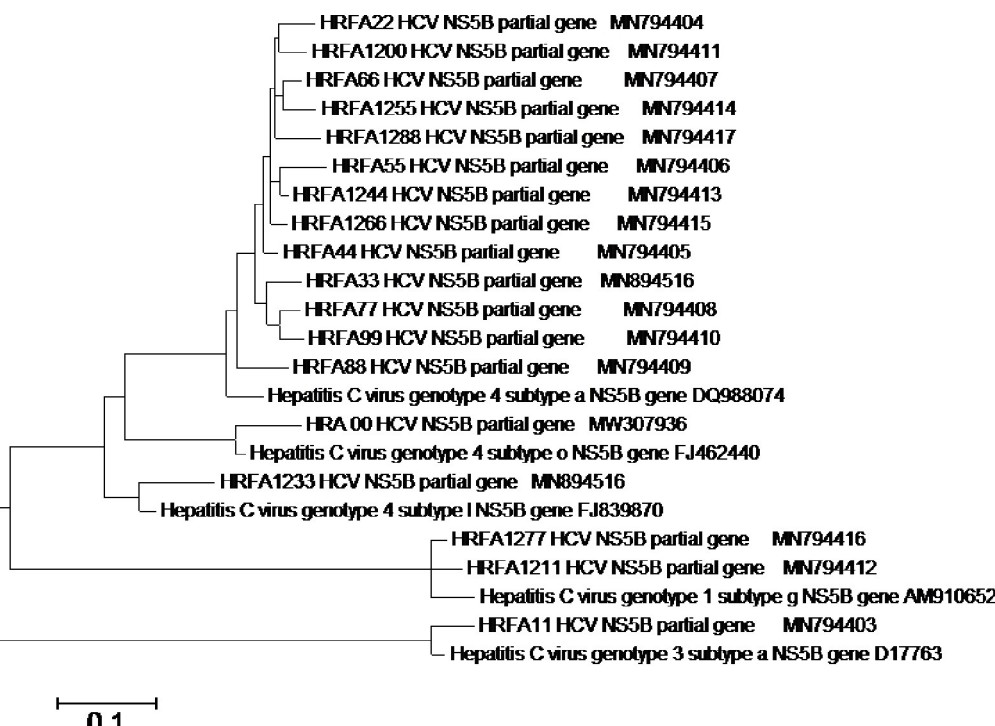

**Fig 1. Molecular phylogenetic analysis by Maximum Likelihood method.** The evolutionary history was inferred by using the Maximum Likelihood method based on the General Time Reversible model [35]. The tree with the highest log likelihood (-2845.8745) is shown. Initial tree(s) for the heuristic search were obtained automatically by applying Neighbor-Join and BioNJ algorithms to a matrix of pairwise distances estimated using the Maximum Composite Likelihood (MCL) approach, and then selecting the topology with superior log likelihood value. A discrete Gamma distribution was used to model evolutionary rate differences among sites (5 categories (+$G$, parameter = 0.2970)). The rate variation model allowed for some sites to be evolutionarily invariable ([+$I$], 0.0000% sites). The tree is drawn to scale, with branch lengths measured in the number of substitutions per site. The analysis involved 23 nucleotide sequences. Codon positions included were 1st. There were a total of 593 positions in the final dataset. Evolutionary analyses were conducted in MEGA6 [36].

combinations of 10 to 15 substitutions (41.1%). However, only one mutation was observed in the successfully amplified sample of a non-responder patient (Table 3).

Table 4 displayed the distribution of amino acids substitution mutations in treatment-naïve patients of 4a genotype. As Combinations of R231K, P300T, K304R, E327D were common between most of the examined treatment-naïve patients.

For samples with multiple mutations, it was detected that specimen with ID 8 found to have 13 substitution mutation sites whose patient of 58 years old representing normal clinical properties. Patient's sample with ID 4 displayed 12 substitution mutation sites presenting abnormal level of ALT only. Patient's specimen with ID 13 revealed 11 substitution mutations sites and found to have abnormal level of ALT only. In addition, variable clinical properties were detected in patients' samples IDs 3, 10, 14, 15 expressing 10 mutations which indicates that no associations between the number of substitutions and the clinical features of the patients.

## 4. Discussion

Although DAA interferon-free treatment displayed a vital development in HCV treatment, the residence of the virus under drug pressure and the residence of natural polymorphism that might be correlated with resistance to DAAs are considered a high challenge to the success in HCV treatment.

**Table 2. Frequency of different HCV NS5B amino acid substitutions mutations among the tested treatment-naïve patients and non-responders.**

| Substitution mutations | No of cases (%) |
|---|---|
| **Treatment-naïve patients (n = 17)** | |
| K304R | 14 (82.4) |
| E327D | 13 (76.5) |
| P300T | 13 (76.5) |
| R270K | 11 (64.7) |
| N333S/R/A | 10 (58.8) |
| R231K | 9 (52.9) |
| V252A | 8 (47) |
| Y285F | 7 (41.2) |
| T254A | 5 (29.4) |
| A235V/T | 4 (23.5) |
| E237G/A | 4 (23.5) |
| M343L/N/I | 3 (17.6) |
| H267Y/R | 3 (17.6) |
| I303T/L | 3 (17.6) |
| K270R | 2 (11.7) |
| L293M | 2 (11.7) |
| I251V | 2 (11.7) |
| R254K | 2 (11.7) |
| K307G/R | 2 (11.7) |
| A231G | 1 (5.8) |
| T235M | 1 (5.8) |
| A252V | 1 (5.8) |
| A255S | 1 (5.8) |
| N255S | 1 (5.8) |
| T257S | 1 (5.8) |
| D272E | 1 (5.8) |
| L273F | 1 (5.8) |
| T286P | 1 (5.8) |
| M300T | 1 (5.8) |
| A302S | 1 (5.8) |
| V285F | 1 (5.8) |
| N307G | 1 (5.8) |
| Q309R | 1 (5.8) |
| R337G | 1 (5.8) |
| A342V | 1 (5.8) |
| T344D | 1 (5.8) |
| R345Q | 1 (5.8) |
| V322I | 1 (5.8) |
| G333A | 1 (5.8) |
| L336I | 1 (5.8) |
| N244D | 1(5.8) |
| D244A | 1 (5.8) |
| **Non-responders (n = 1)** | |
| E237G | 1 (100) |

**Table 3. Number of NS5B substitutions per case.**

| Number of substitutions mutation | Number of cases (%) |
|---|---|
| **Treatment-naïve patients (n = 17)** | |
| less than 5 substitutions | 2 (11.7) |
| [$\geq$ 5 to < 10] substitutions | 8 (47) |
| [$\geq$ 10 to <15] substitutions | 7 (41.1) |
| More than 15 substitutions | 0 (0) |
| **Non-responders (n = 1)** | |
| less than 5 substitutions | 1 (100) |

Our study was done on 27 treatment-naïve patients and 8 non-responders. The non-responders showed low viral load in comparison to treatment-naïve patients. This because the virus was under the stress of the antiviral therapy resulting in decrease in the replication of the virus and low viral load. In addition, many patients under the antiviral therapy may show inaccurate low viral load or false negativity when testing plasma for HCV RNA due to the presence of HCV distributed in hepatic tissues and in extrahepatic tissues such as pancreas, kidney, brain, lymphoid tissues or elsewhere. So, direct testing of these tissues in individuals who test negatively for HCV RNA in blood will be required. These tissue sites could serve as potential reservoirs of persistent HCV infection and a viral relapse source [37–40].

Seventeen specimens of treatment-naïve patients and one specimen of non-responders were successfully amplified for the examined NS5B fragments. To our consideration, this study is one of few studies, which estimated NS5B RASs naturally occurring in DAA-naïve HCV patients in the Egyptian community. In order to provide optimal therapy, especially in non-responders, population surveillance to identify the baseline prevalence of HCV direct resistance mutations (DRMs), such as S282T, is important. Although deep sequencing was used to study NS5B regions, these studies either failed to examine DRMs; restricted by homogeneity of the HCV subtype; or small cohorts of infected individuals were used [41–43].

Concerning NS5B gene, the global analysis revealed that NS5B DAA resistance substitutions are infrequent [44]. In our study relevant natural aa polymorphisms were observed in genotypes 4a, 4l, 3a, 1g and 4o. In the NS5B region, some studies have shown the appearance of naturally occurring mutations. A study from Argentina on 108 HCV-1-infected patients, NS5B RASs were detected in 6.3% [45]. Moreover, another study showed that the prevalence of RASs at baseline was 20.9% in 43 HCV strains underwent extra analysis of the HCV NS5B by next generation sequencing [46]. Despite, an Italian study described that from 45 HCV GT3a patients 0% was harboring known RAS at baseline for NS5B [47]. Another study revealed that out of 10 investigated genotype 1a strains, 5 strains had mutations in codons K304, A327, S254, and N307 [48]. Also, multiple substitutions at the same aa positions were detected among our HCV strains.

Concerning mutations associated with ribavirin resistance at positions Q309R, D244N, and A333E, a study performed in Brazil reported that among 69 drug-naïve individuals with hepatitis C virus (HCV) infections, the most common mutation was Q309R (29%) [49]. These results extend and consolidate previous study which displayed the high prevalence of the Q309R mutation [49]. However, this mutation (Q309R) was reported in one (5.6%) of our cases.

Additionally, a study on 42 HCV sequences recorded that D244N mutation was not detected. A333E mutation was recognized in 11 (50%) sequences of genotype 5a. Other peg-IFN/RBV resistance-associated mutations, D310N and T329I, were not detected in the examined sequences [49]. These findings to some extent resemble our results, as these mutations

**Table 4. Distribution of different amino acids mutations in NS5B in treatment-naïve patients and non-responders of different HCV genotype.**

| Accession No. | ID | Genotype Subtype | No. of substitutions | MUTATIONS | | | | | | | | | | | | | | | | | | | | | | | | | | | | |
|---|---|---|---|---|---|---|---|---|---|---|---|---|---|---|---|---|---|---|---|---|---|---|---|---|---|---|---|---|---|---|---|---|
| | | | | 231 | 235 | 237 | 244 | 251 | 252 | 254 | 255 | 257 | 267 | 270 | 272 | 273 | 285 | 286 | 293 | 300 | 302 | 303 | 304 | 307 | 309 | 322 | 327 | 333 | 336 | 337 | 342 | 343 | 344 | 345 |
| **Treatment-naïve patients** | | | | | | | | | | | | | | | | | | | | | | | | | | | | | | | | | |
| MN794403 | 1 | 3a | 3 | | | | N244D | | | | | | | | | | | | | | | | K304R | N307G | | | | | | | | | | |
| MN889516 | 2 | 4a | 7 | R231K | | E237A | | | | | | | | R270K | | | Y285F | | | P300T | | | K304R | | | | E327D | | | | | | | |
| MN794406 | 3 | 4a | 10 | R231K | | | | | V252A | T254A | | | | R270K | | | | T286P | | P300T | | | K304R | | | V322I | E327D | N333S | | | | | | |
| MN794404 | 4 | 4a | 12 | R231K | | E237G | | | V252A | T254A | A255S | | | R270K | | | | | L293M | P300T | | | K304R | | | | E327D | N333R | | R337G | A342V | M343L | | |
| MN794405 | 5 | 4a | 7 | R231K | | | | | V252A | | | | | R270K | | | Y285F | | | P300T | | | K304R | | | | E327D | N333S | | | | | | |
| MN794407 | 6 | 4a | 7 | R231K | | | | | | | | | | R270K | | | Y285F | | | P300T | | | K304R | | | | E327D | N333S | | | | | | |
| MN794408 | 7 | 4a | 8 | R231K | A235V | | | | V252A | | | | | R270K | | | Y285F | | | P300T | | | K304R | | | | E327D | | | | | | | |
| MN794409 | 8 | 4a | 13 | R231K | A235T | E237G | D244A | | V252A | | | | H267Y | R270K | | L273F | | | | P300T | | I303L | K304R | | | | E327D | N333A | | | | | | |
| MN794410 | 9 | 4a | 7 | R231K | | | | | V252A | T254A | | | | R270K | | | | | L293M | P300T | | | K304R | | | | E327D | | | | | | | |
| MN794411 | 10 | 4a | 10 | R231K | | | | | V252A | T254A | | | | R270K | | | V285F | | | P300T | | | K304R | | | | E327D | N333S | | | | M343I | | |
| MN794412 | 11 | 1g | 6 | A231G | | | | I251V | | R254K | N255S | | | K270R | | | | | | | | | | K307R | | | | | | | | | |
| MN889517 | 12 | 4l | 3 | | | | | | | | | | | | | | | | | M300T | | | | | | | | G333A | | | | | | |
| MN794413 | 13 | 4a | 11 | R231K | A235V | | | | V252A | T254A | | | | R270K | | | Y285F | | | P300T | | | K304R | | | | E327D | N333S | L336I | | | | | |
| MN794414 | 14 | 4a | 10 | R231K | A235V | | | | V252A | | | | H267Y | R270K | | | Y285F | | | P300T | | I303T | K304R | | | | E327D | N333S | | | | | | |
| MN794415 | 15 | 4a | 10 | R231K | | E237G | | | | R254K | | | | R270K | | | | | | P300T | | I303T | K304R | | | | E327D | N333S | | | | | | |
| MN794416 | 16 | 1g | 8 | | T235M | | | I251V | A252V | R254K | | T257S | H267Y | | | | | | | | A302S | | | K307G | Q309R | | | | | | | M343N | T344D | R345Q |
| MN794417 | 17 | 4a | 9 | | A235T | | | | A252V | | | | H267R | R270K | D272I | | Y285F | | | P300T | | | K304R | | | | E327D | N333S | | | | | | |
| **Non-responders** | | | | | | | | | | | | | | | | | | | | | | | | | | | | | | | | | |
| MW307936 | 28 | 4o | 1 | | | E237D | | | | | | | | | | | | | | | | | | | | | | | | | | | | | |

were not recognized in our strains. Whereas N333S/R/A polymorphisms were identified in 10 of 18 strains (55.6%), both D244A and N244D polymorphisms were recognized in only one strain (5.6%).

Another study found that the resistant mutation (A300T) showed high distribution between genotype 1b strains (82.7%) [50]. Additionally, a study performed on 16 naive HCV/HIV-1 coinfected patients, recorded substitutions on amino acid at position 300 (A300T, A300S, R300Q) [41]. Among our strain's multiple substitutions, P300T (72.2%) and M300T (5.6%), were identified at the same position.

Significant S282T mutation that induce a high level of SOF resistance as well as other recorded mutations that are clinically significant in SOF resistance (L320F/C) and (C316Y/N) were not recognized in any of our examined sequences. These findings agree with those observed in other studies which found that the S282T mutation show low prevalence among HCV patients [41, 42]. Furthermore, larger studies concerning HCV patients with genotypes 1, 2, and 3 Failed to detect the S282T mutation between treatment-naïve and treatment-experienced individuals [29, 51–54]. Also, many studies displayed that the emergence of S282T substitution was limited in patients who fail SOF-based regimens [55, 56]. Furthermore, S282T substitution was not reported in a study performed on 127 HCV-positive specimens of the Canadian cohort while S282G/C/R substitutions were recognized in nine patients [57]. Another study reported that 5 patients (38.5%) showing treatment failure harbored NS5B S282C/T RASs [58].

Our study reported the NS5B RAS E237G in the only non-responsive HCV GT-4o strain and E237G/A mutations in another 4 strains of our naive HCV specimens. This outcome agrees to some extend with a study on 333 patients who received therapy, ten patients had a virological relapse, E237G mutation was detected in two patients with genotype 1a and genotype 4d at the time of relapse [59]. Another study recorded the presence of E237G substitution in one patient of GT- 4 with no significant therapeutic response [60].

Some novel baseline NS5B polymorphisms may not influence therapy outcomes, as they have not been enriched in post-failure specimens. While baseline resistant variants might cause viral breakthroughs during therapy. The effect of HCV-resistance mutations on the capacity of the virus to replicate *in-vivo* remains unclear. Additional studies are required for better assessment of the impact of all variants on modulating resistance levels or susceptibility to HCV drugs.

## Study limitations and future recommendations

- **The small number of the analyzed samples, especially the non-responders:** The number of non-responders is very low due to the high rate of virological success of SOF-based regimens. Additionally, many HCV Egyptian patients received the treatment at private clinics and pharmacies outside the viral hepatitis management center so it was hard to reach them to be included in our study.

- **Lack of comparison with other cohort of patients who achieved an early viral response:** There were difficulties following up the patients after the end of treatment. We tried to follow up our cases responsiveness to the treatment but failed. In Egypt, the routine follow-up depends mainly on performing RT-PCR to detect the viral load every 6 months, then every year. So, patients can perform this test in private laboratories outside the viral hepatitis management center.

- **The use of multiple DAAs but analyzing one target gene.** Although daclatasvir targets NS5A, nothing was done on this gene as patients may be resistant to daclatasvir and this

resistance was related to NS5A and not NS5B. This is because the sequencing step has a high cost, and this research did not receive any specific grant from funding agencies.

- To undergo the alignment step, HCV GT-4a reference strain with accession number DQ988074, HCV GT-4l reference strain FJ839870, HCV GT-4o reference strain FJ462440, HCV GT-3a reference strain D17763 and HCV GT-1g reference strain AM910652 from NCBI were chosen to be aligned with the sequenced samples. As a result, for the alignment, we noticed the presence of the mutations or the substitutions listed in our study in the tested samples and their absence in the reference strains. So, further investigations are needed to prove their relation to resistance.

## 5. Conclusion

Our study reported that there are many combinations of multiple aa substitutions in the analyzed NS5B genes which could enhance therapy failure possibility in HCV patients treated with regimens containing DAA. Overall, it may be difficult to fully understand the role of some of these substitutions, the high virological success rate of SOF-based regimens, the rare incidence of RAS in non-responsive patients, and the minimal data developed for RAS in genotype 4 patients compared to other genotypes.

Further studies are needed to test the relation between the reported substitutions in our study and HCV resistance.

## Author Contributions

**Data curation:** Hala Rady Ahmed, Rehab Mahmoud Abd El-Baky, Ramadan Yahia.

**Methodology:** Hala Rady Ahmed, Helal F. Hetta, Amr M. Elsayed.

**Supervision:** Nancy G. F. M. Waly, Reham Ali Ibrahem.

**Writing – original draft:** Hala Rady Ahmed, Nancy G. F. M. Waly, Rehab Mahmoud Abd El-Baky, Helal F. Hetta, Reham Ali Ibrahem.

**Writing – review & editing:** Nancy G. F. M. Waly, Rehab Mahmoud Abd El-Baky, Helal F. Hetta, Reham Ali Ibrahem.

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
