## [Decision Letter · Decision Letter 0]

1 Feb 2021

PONE-D-20-39695

Distribution of naturally occurring NS5B resistant

associated substitutions among treatment naïve HCV Egyptian patients and non-responders

PLOS ONE

Dear Rehab Mahmoud Abd El-Baky,

Thank you for submitting your manuscript to PLOS ONE. After careful consideration, we feel that it has merit but does not fully meet PLOS ONE’s publication criteria as it currently stands. Therefore, we invite you to submit a revised version of the manuscript that addresses the points raised during the review process.

Although the reviewers found your study relevant, some issues need to be addressed including the small number of analyzed samples, English editing, a too small NS5B region sequenced, to compare sequences to a set of HCV NS5B sequences, etc.

We look forward to receiving your revised manuscript.

Kind regards,

Philippe Gallay

Academic Editor

PLOS ONE

Journal Requirements:

2.Thank you for submitting the above manuscript to PLOS ONE. During our internal evaluation of the manuscript, we found some minor occurrences of overlapping text with the following previous publication(s), some of which you are an author, which needs to be addressed:

- https://www.sciencedirect.com/science/article/abs/pii/S2210740119300439?via%3Dihub

We would like to make you aware that copying extracts from previous publications word-for-word, especially outside the methods section, is unacceptable. In addition, the reproduction of text from published reports has implications for the copyright that may apply to the publications.

Please revise the manuscript to quote or rephrase the duplicated text and cite your sources for text outside the methods section. Please note that further consideration is dependent on the submission of a manuscript that addresses these concerns about the overlap in text with published work.

Reviewers' comments:

Reviewer's Responses to Questions

**Comments to the Author**

1. Is the manuscript technically sound, and do the data support the conclusions?

Reviewer #1: Partly

Reviewer #2: Partly

2. Has the statistical analysis been performed appropriately and rigorously? 

Reviewer #1: Yes

Reviewer #2: N/A

3. Have the authors made all data underlying the findings in their manuscript fully available?

Reviewer #1: Yes

Reviewer #2: Yes

4. Is the manuscript presented in an intelligible fashion and written in standard English?

Reviewer #1: No

Reviewer #2: No

5. Review Comments to the Author

Reviewer #1: This is a cross-sectional study of NS5B resistance mutations in Egypt. Given the very high prevalence of HCV in this part of the world, such studies are relevant. However, there are a number of significant limitations to this study.

Most importantly, the very small sample size is quite surprising, since there are clearly many more individuals that could have been included / studied.

The manuscript requires editing by a native English speaker and/or professional editing service.

How was HCV RNA quantified? What was the lower limit of detection for the assay?

Only 18 of the 35 patient samples could be amplified by PCR suggesting very low sample quality. Samples storage conditions should be stated explicitly.

Were the samples that were PCR negative then extracted again and the PCR tried a second time?

The authors have amplified only a small portion (~400 bases) of the entire NS5B region, representing a significant study limitation.

“normal” and “abnormal” levels for ALT and other clinical variables should be explicitly defined.

Are the authors comparing their sequences to a single NS5B consensus to identify mutations? If so, this approach has major limitations. The consensus may change with more or fewer sequences to generate the consensus, and minor – but important – mutations may be missed altogether. It is better to compare the query sequence to a set of HCV references (rather than to one) and then define a mutation as greater than X percentage of samples.

A phylogenetic tree must be included to show the relationship amongst the sequences isolates.

Reviewer #2: In this manuscript authors analyzed the RASs occurrence in 17 treatment-naïve and 1 non-responder samples out of 35 total number of patients. Samples were collected from patients who failed to achieve an early viral response with combined sofosbuvir (400mg/day), daclatasvir (60mg/day) for 6 months therapy. Authors reported several mutations as possible RASs linked to SOF.

Although these mutations may be of a great value to understand DAA resistance, however, the study lack important aspects that need to be addressed to be publishable. Examples for these shortcomings are the small number of the analyzed samples especially the non-responders, lack of comparison with other cohort of patients who achieved an early viral response, and the use of multiple DAAs but analyzing one target gene. Although daclatasvir targets NS5A, nothing was done on this gene as patients may be resistant to daclatasvir and this resistance was related to NS5A and not NS5B. The manuscript has many places that need rephrasing or improvement to read better. In addition, the references need to be updated as the authors did not include several important and recent papers about the topic.

More detailed comments will be attached below.

6. PLOS authors have the option to publish the peer review history of their article (what does this mean?). If published, this will include your full peer review and any attached files.

Reviewer #1: No

Reviewer #2: No

---

## [Author Response · Author response to Decision Letter 0]

27 Feb 2021

Reviewers’ comments Author’s replies

The manuscript requires editing by a native English speaker and/or professional editing service. The authors made the required English editing to the manuscript.

How was HCV RNA quantified? What was the lower limit of detection for the assay?

 The HCV-RNA was amplified on Stratagene Mx3000P system using Artus HCV QS-RGQ-PCR Kit reagents. Amplification followed by simultaneous detection was carried out using RT-PCR, with TaqMan assay for a particular region of the HCV genome. the lower limit of detection for the assay is 30 IU/ml.

Only 18 of the 35 patient samples could be amplified by PCR suggesting very low sample quality. Samples storage conditions should be stated explicitly.

 Approximately, 3 ml of blood were collected from each patient, the samples were shipped cooled (4°C) within 6 hours. centrifuged at 1500 rpm, sera were separated then distributed in aliquots, stored at -20°C. Then after RNA extraction the eluted RNA was stored at -80°C until further use. The sensitivity of the assay can be reduced as the samples stored for a longer period of time. 

The assay may be subjected to many issues such as difficulty to amplify NS5B fragments, low viral load and the need to extremely trained professionals. These issues were discussed in the following references:

• Abdel-Hamid M, El-Daly M, Molnegren V, El-Kafrawy S, Abdel-Latif S, Esmat G, et al. Genetic diversity in hepatitis C virus in Egypt and possible association with hepatocellular carcinoma. Journal of general virology. 2007;88(5):1526-31.

• Li Z, Zhang Y, Liu Y, Shao X, Luo Q, Cai Q, et al. Naturally occurring drug resistance associated variants to hepatitis C virus direct-acting antiviral agents in treatment-naive HCV genotype 1b-infected patients in China. Medicine. 2017;96(19).

• Warkad SD, Nimse SB, Song K-S, Kim T. HCV detection, discrimination, and genotyping technologies. Sensors. 2018;18(10):3423.

Were the samples that were PCR negative then extracted again and the PCR tried a second time?

 Yes, the samples that were PCR negative extracted again using The MagMAX Viral RNA Isolation Kit For viral nucleic acid isolation which used to deliver yields of high quality RNA and DNA with little sample-to-sample variation as 35–75% of the input RNA should be recovered. The MagMAX™ magnetic bead technology useful for concentrating RNA from dilute samples. Then, the PCR tried a second time but gave no bands.

The authors have amplified only a small portion (~400 bases) of the entire NS5B region, representing a significant study limitation.

 Because Sofosbuvir (SOF) is the predominant available NS5B polymerase inhibitor Nucleoside/nucleotide inhibitors (NI)] in Egypt, we mainly investigated the positions which are related to resistance associated variants (RAVs) to sofosbuvir and other positions such as 368, 411, 414, 448, 553, 554 and 556 which are related to resistance to another NS5B polymerase inhibitor, Dasabuvir [non-nucleoside inhibitor] NNI)], not fully covered in this study. As, NNIs delivered as a monotherapy came with a limited success, with a low to average antiviral activity, a low resistance barrier, and a minimal range of genotypic activity (active only against subtypes 1b and 1a).

Fadl, N, Salem, TZ. Hepatitis C genotype 4: A report on resistance‐associated substitutions in NS3, NS5A, and NS5B genes. Rev Med Virol. 2020; 30:e2120

“normal” and “abnormal” levels for ALT and other clinical variables should be explicitly defined. Normal and abnormal levels for all clinical variables defined in the manuscript.

Are the authors comparing their sequences to a single NS5B consensus to identify mutations? If so, this approach has major limitations. The consensus may change with more or fewer sequences to generate the consensus, and minor – but important – mutations may be missed altogether. It is better to compare the query sequence to a set of HCV references (rather than to one) and then define a mutation as greater than X percentage of samples.

 In order to undergo alignment step, HCV GT-4a reference strain with accession number DQ988074, HCV GT-4l reference strain FJ839870, HCV GT-4o reference strain FJ462440, HCV GT-3a reference strain D17763 and HCV GT-1g reference strain AM910652 from NCBI (National Center for Biotechnology Information) was chosen to be aligned with the sequenced samples.

As a result, for the alignment, we noticed the presence of the mutations or the substitutions listed in our study in the tested samples and their absence in the reference strains. So, further investigations are needed to prove their relation to resistance.

A phylogenetic tree must be included to show the relationship amongst the sequences isolates.

 The authors added the phylogenetic tree to the manuscript.

The small number of the analyzed samples, especially the non-responders.

.

 Most of HCV Egyptian patients received the treatment at private clinics and pharmacies outside the viral hepatitis management center so hardly to reach to be included in our study. Additionally, the number of non-responder cases are very low due to the high rate of virological success of SOF-based regimens.

lack of comparison with other cohort of patients who achieved an early viral response.

 There is some difficulty to follow up the patients after they end their treatment. We already tried to follow up our cases responsiveness to the treatment but failed. In Egypt, the routine follow-up depends mainly on performing RT-PCR to detect the viral load every 6 months, Then every year. So, patients can perform this test in private laboratories not the center which is considered one of the limitations of this study.

the use of multiple DAAs but analyzing one target gene. Although daclatasvir targets NS5A, nothing was done on this gene as patients may be resistant to daclatasvir and this resistance was related to NS5A and not NS5B.

.

 Because sequencing step has high cost and this research did not receive any specific grant from funding agencies.

The manuscript has many places that need rephrasing or improvement to read better The authors made the required rephrasing to the manuscript.

In addition, the references need to be updated as the authors did not include several important and recent papers about the topic. The authors made the required updates to the references.

In the paragraph started with “Ribavirin-associated resistance mutations at positions D244N, Q309R, and A333E of NS5B were analyzed.” I am not sure why the authors compare their mutations with non-DAAs drugs and ignore comparing them with other DAAs drugs. Either they include the most related drugs or remove this paragraph altogether.

 Ribavirin is included in HCV treatment protocol in Egypt. RBV is a nucleotide analogue that can be incorporated in the RNA-dependent polymerase, although the precise mechanism of action on viral replication remains elusive.

Raj, V. S., Hundie, G. B., Schürch, A. C., Smits, S. L., Pas, S. D., Le Pogam, S., ... & Haagmans, B. L. (2017). Identification of HCV resistant variants against direct acting antivirals in plasma and liver of treatment naïve patients. Scientific reports, 7(1), 1-10.

‏

Petronella Gededzha, M., Mphahlele, M. J., Blackard, J. T., & Gloria Selabe, S. (2017). Prevalence of NS5B resistance mutations in hepatitis C virus (HCV) treatment naïve South Africans. Hepatitis Monthly, 17(6).‏ 

Also, in Table 1, non-responders showed low viral load in comparison to naïve patients. This has to be addressed and reasonable explanation has to be made.

 As the virus was under the stress of the antiviral therapy which results in the decrease in the replication of the virus and low viral load while naïve treatment patients are not subjected to the antiviral therapy stress. In addition, many patients under the antiviral therapy showed inaccurate low viral load or false negativity when testing plasma for HCV RNA due to the presence of HCV distributed in hepatic tissues and in extrahepatic tissues such as pancreas, kidney, brain, lymphoid tissues or elsewhere. So, Direct testing of

these tissues in individuals who test negatively for

HCV RNA in blood will be required to confirm or refute

that these other tissue sites serve as potential reservoirs

of persistent HCV infection and as a source of

viral relapse.

- Gurakar A, Fagiuoli S, Faruki H, De Maria N, Balkan M, Van Thiel DH, Friedlander L. Utility of hepatitis C virus RNA determinations in hepatic tissue as an end point for interferon treatment of chronic hepatitis C. Hepatology. 1995 Oct;22(4 Pt 1):1109-12. doi: 10.1016/0270-9139(95)90616-9. PMID: 7557858.

Change all treatment naïve to “treatment-naïve” The authors change all treatment naïve to “treatment-naïve” in the manuscript

Title

- For “Distribution of naturally occurring NS5B resistant associated substitutions among treatment- naïve HCV Egyptian patients and non-responders”

The title is not accurate as the authors should not consider the non-responders data as it came out from only one patient. We suggested this title

Distribution of naturally NS5B resistant associated substitutions between Egyptian HCV in Minia, Egypt

Abstract

- Authors mentioned “Results: NS5B amino acid sequence analysis showed the presence of several novel NS5B mutations (more than 40 substitution mutations) that have not been previously reported to be associated with a resistant phenotype.”

Without comparing with another cohort with EVR, these mutations may be not related to resistance. We compare our results with 5 standard strains sequences. we noticed the presence of the substitutions that were listed in our study in the tested samples and their absence in the reference strains and we did not find these substitutions in any of the research that were carried out in our area. So, further investigations are needed to prove their relation to resistance.

- In Table 4, it is not clear why the authors arrange ID numbers or the mutations this way. Neither IDs, number of mutations, or location of the mutations are in order. I recommend to order the table according to the no. of substitutions as well as the locations of the mutations. Each column should include one location and if at that position different mutations exist, it should be market with asterisk. Example, first column should be only for position 231 with asterisk for A231G* and moving 244 to different column in order. All changes were done

Note: the alignment for the substitutions were put in separate columns but the table became too large. So, we add it in the manuscript as an image. The actual excel form of the table will be attached as supplementary file.

All typing errors written in the response file of the reviewer were corrected in the manuscript.

---

## [Editor Report · Decision Letter 1]

8 Mar 2021

PONE-D-20-39695R1

Distribution of naturally occurring NS5B resistant associated substitutions among treatment naïve HCV Egyptian patients and non-responders

PLOS ONE

Dear Dr. Rehab Mahmoud Abd El-Baky,

Thank you for submitting your manuscript to PLOS ONE. After careful consideration, we feel that it has merit but does not fully meet PLOS ONE’s publication criteria as it currently stands. Therefore, we invite you to submit a revised version of the manuscript that addresses the points raised during the review process.

The authors have addressed a majority of the previous issues. However, their explanation/comments need to be fully integrated in the text starting with the title "....treatment-naive".

We look forward to receiving your revised manuscript.

Kind regards,

Philippe Gallay

Academic Editor

PLOS ONE

Journal Requirements:

Additional Editor Comments (if provided):

The authors addressed a majority of the issues and provided reasonable answers to address the comments of the reviewers. However, all their explanation need to be integrated in the revised manuscript starting with the title: "....treatment-naive.....".

---

## [Author Response · Author response to Decision Letter 1]

23 Mar 2021

Authors respond to all reviewers' comments and the file was attached

---

## [Editor Report · Decision Letter 2]

25 Mar 2021

Distribution of Naturally -Occurring NS5B Resistance-Associated Substitutions In Egyptian Patients with Chronic Hepatitis C

PONE-D-20-39695R2

Dear Dr. Rehab Mahmoud Abd El-Baky,

We’re pleased to inform you that your manuscript has been judged scientifically suitable for publication and will be formally accepted for publication once it meets all outstanding technical requirements.

Kind regards,

Philippe Gallay

Academic Editor

PLOS ONE
---

## [Editor Report · Acceptance letter]

6 Apr 2021

PONE-D-20-39695R2 

Distribution of Naturally -Occurring NS5B Resistance-Associated Substitutions In Egyptian Patients with Chronic Hepatitis C

Dear Dr. Abd El-Baky:

I'm pleased to inform you that your manuscript has been deemed suitable for publication in PLOS ONE. Congratulations! Your manuscript is now with our production department. 

Kind regards, 

on behalf of

Prof. Philippe Gallay 

Academic Editor

PLOS ONE